# Quantum criticality at cryogenic melting of polar bubble lattices

Wei Luo [1], Alireza Akbarzadeh[1,2], Yousra Nahas [1], Sergei Prokhorenko [1] ✉ & Laurent Bellaiche [1] ✉

Quantum fluctuations (QFs) caused by zero-point phonon vibrations (ZPPVs) are known to prevent the occurrence of polar phases in bulk incipient ferroelectrics down to 0 K. On the other hand, little is known about the effects of QFs on the recently discovered topological patterns in ferroelectric nanostructures. Here, by using an atomistic effective Hamiltonian within classical Monte Carlo (CMC) and path integral quantum Monte Carlo (PI-QMC), we unveil how QFs affect the topology of several dipolar phases in ultrathin $Pb(Zr_{0.4}Ti_{0.6})O_3$ (PZT) films. In particular, our PI-QMC simulations show that the ZPPVs do not suppress polar patterns but rather stabilize the labyrinth, bimeron and bubble phases within a wider range of bias field magnitudes. Moreover, we reveal that quantum fluctuations induce a quantum critical point (QCP) separating a hexagonal bubble lattice from a liquid-like state characterized by spontaneous motion, creation and annihilation of polar bubbles at cryogenic temperatures. Finally, we show that the discovered quantum melting is associated with anomalous physical response, as, e.g., demonstrated by a negative longitudinal piezoelectric coefficient.

Owing to the small energy difference between different distorted-perovskite structures, quantum fluctuations (QFs) can play an important role in the ferroelectric phase transitions of perovskite materials. One typical example is the bulk $SrTiO_3$. The QFs caused by the zero-point phonon vibrations (ZPPVs) there prevent a ferroelectric phase transition in this material, therefore freezing the system into a macroscopic paraelectric structure down to 0 K (see, e.g., refs. 1,2 and references therein). Similar effects due to QFs were also found in other perovskites, such as bulk $BaTiO_3$ under hydrostatic pressure[3,4] and $KTaO_3$ bulk and nanodots[5–7].

At the same time, the role of QFs in atomically thin ferroelectrics is currently unknown, which is unfortunate in the view of the recent discoveries of various technologically prominent topological domain patterns in ultra-thin ferroelectric layers and films. The latter include polar vortices also known as nanostripes[8,9], skyrmions[10–13], bubbles[14,15] and skyrmion bubbles[16–18], merons[19], dipolar waves[16] and labyrinths[20]. These phases are deemed promising for enhanced or novel applications and phenomena, including high storage density[21], switchable

conductivity[22], negative capacitance[23–25] and negative piezoelectric effect[26]. Nonetheless, to the best of our knowledge, we are not aware of any study investigating the effects of QFs on topological patterns in ultrathin ferroelectrics.

One may for instance wonder if QFs can quantitatively but also qualitatively affect the various low-temperature topological phases recently predicted by classical Monte Carlo (CMC) simulations in quenched $Pb(Zr_{0.4}Ti_{0.6})O_3$ (PZT) films[18]. The latter comprise the so-called connected polar labyrinths that evolve into disconnected labyrinthine patterns, mixed bimerons-bubbles, bubbles and then monodomain when increasing the magnitude of the bias electric field (note that these phases have also been observed by experiments[18,20]). Will ZPPVs weaken such states as in the case of bulk ferroelectrics or rather unveil novel quantum phases and phenomena? Could QFs lead to anomalous or unusual technological properties? Answering these questions would not only deepen our current understanding of QFs in low-dimensional materials and topology but may also lead to novel technologies. For instance, the discovery of correlated quantum

[1]Physics Department and Institute for Nanoscience and Engineering, University of Arkansas, Fayetteville, AR 72701, USA. [2]Science, Engineering, and Geosciences, Lonestar College, 9191 Barker Cypress Road, Cypress, TX 77433, USA. ✉e-mail: prokhorenko.s@gmail.com; laurent@uark.edu

phases in low-dimensional polar systems could enable new opportunities for quantum[27] and neuromorphic computing[28].

In this work, we address these questions by performing atomistic simulations based on the effective Hamiltonian model used in Ref. 29 both within the Classical Monte-Carlo (CMC) and path-integral quantum Monte Carlo (PI-QMC) scheme[6,7,30,31]. Note that the former approach neglects ZPPVs while the latter incorporates the aforementioned quantum effects. As we will see, surprises are in store since QFs are found to not only quantitatively but also qualitatively affect topological phases and even result in intriguing physical phenomena, such as the creation of a quantum critical point (QCP) and negative longitudinal piezoelectric coefficients.

## Results

The studied PZT films have a thickness of about 2 nm (which corresponds to 5 unit cells) and are subject to a compressive strain of −2.65%, in order to mimic the growth on a SrTiO₃ substrate. The simulation supercell is chosen to be $26 \times 26 \times 5$, and the electrical boundary condition is set to screen 80% of the polarization-induced surface charges (We also discuss the effects of different screening factors, see Supplementary Note 1), in order to be realistic. Within our PI-QMC simulations, the ultrathin films PZT are first thermalized at 500 K with a so-called Trotter number $P = 1, 2, 4, 8, 16$ and 32 and then thermally quenched (that is, rapidly cooled. The quench rate is roughly 10 K/fs or faster) to 20 K (as detailed in the Method Section, $P$ quantifies ZPPV). The scheme is repeated under different magnitudes of the applied dc electric field with a fixed $P$ value during the quench.

### Topological patterns under different Trotter numbers under zero electric field

As consistent with the work of Nahas et al.[18], the resulting pattern with $P = 1$ (which corresponds to the CMC case) is the kinetically arrested polar labyrinth phase (Fig. 1a) that can be described as polar vortex tubes or nanostripes meandering within the films plane[14,15]. In-line with previous studies of QFs in bulk ferroelectrics[4,6,32] we find that progressively switching on QFs (that is, increasing $P = 1$ to $P = 8$) results in a progressively decreasing magnitude of local electric dipoles (Fig. 1a–f). Interestingly, further increasing the Trotter number also yields a reduction of the domain size and fuzzier edges (vortex lines[14,15]) between neighboring domains (as seen by, e.g., comparing the data for $P = 1$ versus $P = 32$ in Fig. 1). Note that the different patterns shown in

Fig. 1e, f for $P = 16$ and $P = 32$ do not mean that the results have not converged with $P$, but rather that many different labyrinthine patterns with almost degenerated energies exist[20]. The criterion for convergence is, in fact, that the Betti number (to be discussed later) is merely unaffected by $P$.

### Phase diagram for ultrathin films PZT from CMC and PI-QMC simulations under electric field

We now turn to the role of QFs under a finite external dc electric field $E_z$ applied along the out-of-plane direction. The range of the bias magnitudes is chosen to be from $0 \times 10^7$ V/m to $150 \times 10^7$ V/m, by steps of $2 \times 10^7$ V/m. Within this set of simulations, we consider only two cases. Namely, the $P = 1$ (CMC case) and $P = 32$ (converged PI-QMC). In both cases, the system is thermally quenched from a thermalized state at 500 K to 20 K, while maintaining the chosen $P$ value. The different phases obtained at 20 K for the PI-QMC simulations with $P = 32$ are indicated as a function of the dc electric field in Fig. 2b with the corresponding dipolar configurations within the middle (001) plane of the film shown in Fig. 2b1-b8. The CMC data is displayed in Fig. 2a and 2a1-2a5, respectively. Note that in the classical case, we find a perfect agreement with the (classical) results obtained in ref. 18. From Fig. 2a, b, one can see that, for the smallest investigated fields, the labyrinths (Phase I in ref. 18) and disconnected labyrinths (Phase II in Ref. 18) predicted by CMC (Fig. 2a1, a2) persist despite the ZPPVs (Fig. 2b1, b2). However, once again, both the dipole magnitudes and the size of domains decrease when QFs are taken into account. With further increasing the strength of the electric field, mixed bimerons-bubbles (Phase III, bimerons-bubbles can be clearly seen from Supplementary Fig. 2), hexagonal bubble lattice[33] (Phase IV) and monodomain (Phase V, in which all dipoles are aligned along the applied electric field) phases sequentially appear for CMC simulations, as shown in Fig. 2a and displayed in Fig. 2a3-a5. Electric field-induced topological phase transition were also reported in other ferroelectric superlattices from classical simulations[34–36].

These three phases still occur for PI-QMC simulations, as revealed in Fig. 2b, 2b4-b5 and 2b8. At the same time, one can note that all phase boundaries are right-shifted towards higher bias magnitudes in the PI-QMC case. Moreover, the window of stability of, e.g., the bubble lattice (Phase IV) in PI-QMC simulations is broadened. Particularly, quantum effects allow the formation of polar bubble lattice within the bias fields' interval of $42 \times 10^7$ V/m in PI-QMC versus $22 \times 10^7$ V/m in CMC.

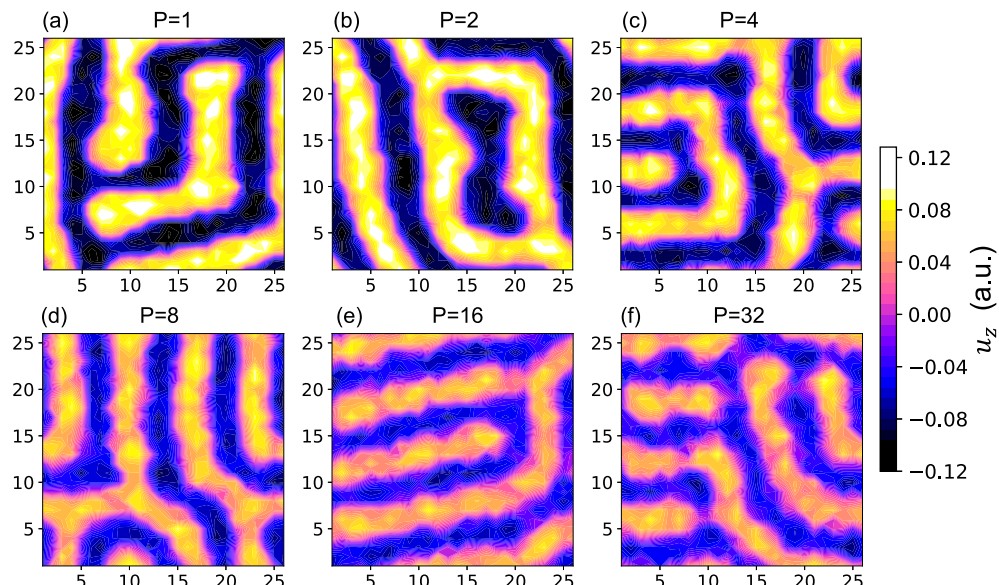

**Fig. 1 | Topological patterns for different Trotter Numbers.** (**a**) $P = 1$, (**b**) $P = 2$, (**c**) $P = 4$, (**d**), $P = 8$, (**e**) $P = 16$ and (**f**) $P = 32$ for PZT films under zero electric field. The $u_z$ represents local modes.

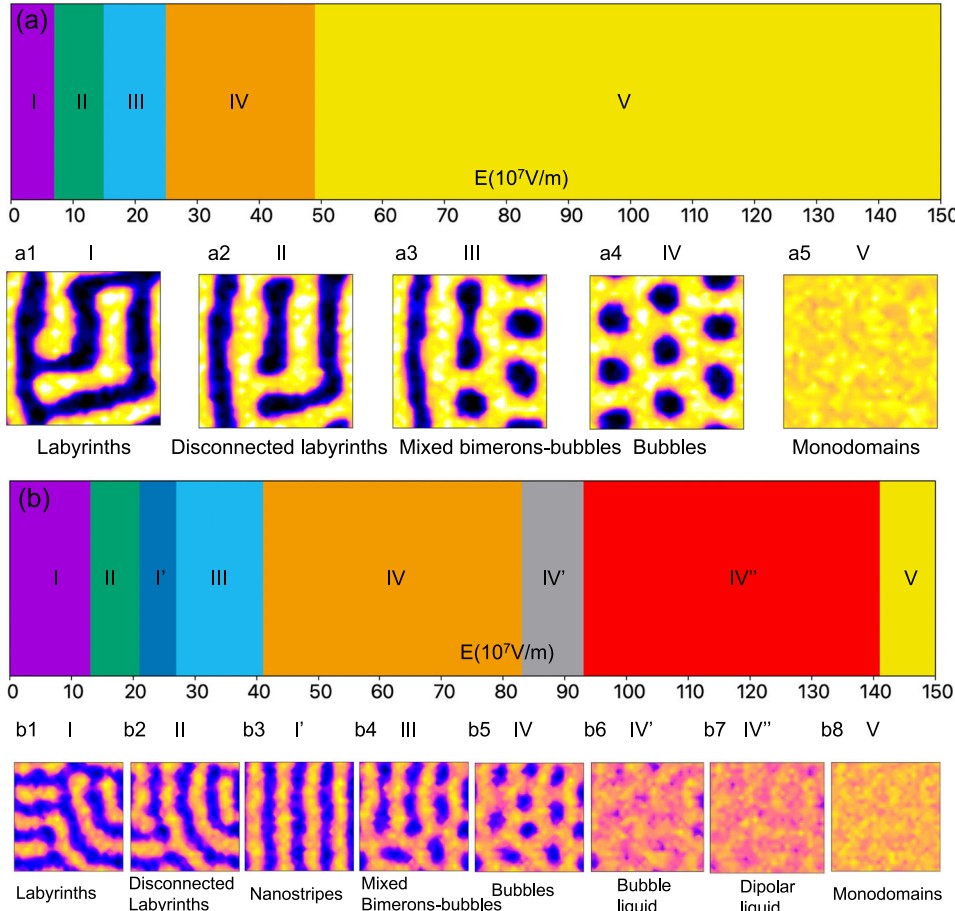

**Fig. 2 | Phase diagram of PZT films as a function of electric fields. a** Phase diagram of PZT films as a function of electric field from CMC calculations ($P=1$). **a1-a5** Selected topological patterns from the last configuration of CMC simulations for Phases I, II, III, IV and V in the middle (001) layer of a $26 \times 26 \times 5$ supercell. (**b**) Same as panel (**a**) but from PI-QMC calculations ($P=32$). **b1-b8** Selected topological patterns from PI-QMC simulations for Phases I, II, I', III, IV, IV', IV" and V in the middle (001) layer of a $26 \times 26 \times 5$ supercell. The yellow (blue) color indicates dipoles that are aligned along the [001] ([00$\bar{1}$]) pseudo-cubic direction.

In addition to the enhanced stability of the bubble lattice phase, ZPPVs also allow for new phenomena and phases. The first emergent phase bridges the disconnected labyrinth and mixed bimeron-bubble patterns. This state is stabilized within the electric field range from $22 \times 10^7$ V/m to $26 \times 10^7$ V/m and is hereon denoted as Phase I'. The corresponding dipolar structure (Fig. 2b3) closely resembles that of the parallel nanostripes which happens to be the ground state of the system at zero field. However, in contrast to the latter, Phase I' possesses a non-zero polarization, with the positive/up stripes having a larger width than the negative/down stripes. Moreover, these stripes change their positions over Monte-Carlo sweeps (see Supplementary Video 3) in phase I', which can therefore also be coined as a dynamic stripe phase.

As surprising as it might appear, this finding is not coincidental. For example, having performed thirty independent quantum quench simulations initialized with different random seeds, we always found the dynamic stripe structure within the indicated field range. Similarly, within the window of stability of Phase II, all independent quantum quenches yielded disconnected polar labyrinths. In contrast, at zero bias field, 47% of the quenches resulted in polar labyrinths and 53% in dynamic stripes. Such results indicate that ZPPVs allow the system to overcome the energy barriers between different realizations of disconnected labyrinths and the dynamic stripes for some electric field magnitudes. This leads to a field-induced straightening of disconnected labyrinths prior to the transition to the bimeron state. This phenomena is analogous to the priorly predicted inverse transition[20] with external electric field playing the role of the temperature.

The second emergent phase exists within the electric field range from $84 \times 10^7$ V/m to $94 \times 10^7$ V/m, and is denoted as Phase IV'. This transient state occurs at the boundary between the bubble lattice and the homogeneously polarized state and, much like Phase IV, also consists of polar bubbles. However, this time, the bubbles have a smaller radius and, unlike domains within Phases I, I', II, III and IV, are dynamic in nature since they change their positions over Monte-Carlo sweeps (see Supplementary Video 1).

Such quantum dynamics of bubbles is an important feature as it drastically changes both the character and the microscopic mechanism of the transition from the bubble lattice to a homogeneously polarized state. Indeed, with no account for ZPPVs, an increasing external field at 20 K triggers a discontinuous transition from the bubble lattice or glass to a homogeneously polarized state (Supplementary Fig. 4). The first-order character of this phase transformation is further stressed by the hysteretic behavior of the polarization with applied field[5] and existence of multiple metastable and static states of depleted bubble arrays[4,5]. Neither of these features are seen here when ZPPVs are taken into account. Moreover, and as will be discussed later, the discontinuous classical behavior of macroscopic observables at the transition becomes continuous when ZPPVs are "switched on". These observations suggest that the quantum transition from the bubble lattice to a homogeneously polarized state acquires a continuous second-order character extended over a finite range of bias magnitudes as QFs allow for dynamical hopping between different local minima of the free-energy landscape. In other words, our PI-QMC results suggest that the motion, as well as the dynamic creation and

annihilation of polar bubbles within Phase IV′ can be described as critical fluctuations triggering a quantum phase transition.

Notably, the strong dipolar fluctuations due to ZPPVs persist even at higher field magnitudes resulting in a yet another structural state. Such final emergent Phase IV″ is shown in Fig. 2b7, and occurs when the electric field ranges from $94 \times 10^7$ V/m to $142 \times 10^7$ V/m (within this range of electric field, CMC simulations give mono-domains. All topological patterns for different electric fields and Trotter number can be found in Supplementary Note 4). We term this state as "dipolar liquid" because it bears resemblance with a state consisting of dipoles being randomly distributed in a crystal[37,38]. Interestingly and as shown in Supplementary Note 5 of the SI (from $E_z = 94 \times 10^7$ V/m to $E_z = 142 \times 10^7$ V/m), its structure factor features a ring-shaped intensity distribution, as representative of dipolar moments being distributed in an isotropic fashion in the real space (note that the structure factors for all topological patterns can be found in Supplementary Fig. 5). Such unusual state also possesses a dynamical character (see Supplementary Video 2), indicating that small barriers between different patterns exist and can be overcome by QFs.

As mentioned above, one can distinguish all the phases summarized in Fig. 2a, b thanks to the computation of their Betti number and their structure factors[18] (see following discussions and Figs S3 and S4 of the SI). As a matter of fact, the Betti number is a topological invariant allowing to distinguish between different morphologies, and is independent of the size, boundary, and shape of domains[39]. Here and in order to differentiate our encountered various states, we just determine the zeroth Betti number $\beta_0$ (i.e., the density of domains), by averaging it over the last 50,000 MC steps, along with the visual inspection of the structure. The evolution of $\beta_0$ as a function of the electric field is shown in Fig. 3a for $P = 1$ (CMC) and Fig. 3b for $P = 32$ (PI-QMC) – while the $\beta_0$ for all our investigated values of $P$ can be found in Supplementary Note 6 (One also can see the convergence trend as a function of Trotter number). The connected labyrinths for $P = 1$ (Phase I, Fig. 2a1) and $P = 32$ (Phase I, Fig. 2b1) have nearly vanishing $\beta_0$, and basically differ by the width of their domains. Phase II for $P = 1$ (Fig. 2a2) and $P = 32$ (Fig. 2b2) exhibits a small but non-vanishing $\beta_0$, as result of the disconnection of some of the labyrinths. For Phase I′ (see Fig. 2b3), $\beta_0$ is independent of the electric field since the PI-QMC simulations always give dynamic stripes with four domains within the chosen supercell. The mixed bimerons-bubbles constituting Phase III possess a Betti number that

typically progressively increases with the field both classically and quantum mechanically. Phase IV (bubble lattice) possesses a significant $\beta_0$ that is basically independent of the electric field, and adopts a value of about 0.0133 for the CMC case versus about 0.0181 for the quantum case for electric fields (such increase in Betti number from the classical to the quantum situation characterizes a larger number of bubbles within the supercell in the latter case). For PI-QMC, the Betti number then adopts a rather different behavior for fields ranging between $84 \times 10^7$ V/m and $142 \times 10^7$ V/m. it increases first from $84 \times 10^7$ V/m to $94 \times 10^7$ V/m, and then decreases from $94 \times 10^7$ V/m to $142 \times 10^7$ V/m. This is because the QFs first decrease the size of the bubbles and make them more numerous between $84 \times 10^7$ V/m to $94 \times 10^7$ V/m (corresponding to the bubble liquid, Phase IV′), but then progressively destroy such bubbles when the electric field varies between $94 \times 10^7$ V/m to $142 \times 10^7$ V/m (corresponding dipolar liquid, Phase IV″). For Phase V, corresponding to a monodomain state with all dipolar moments oriented along the direction of the external dc electric field, $\beta_0$ vanishes again both for CMC and PI-QMC cases. It also remarkable that Fig. 3a further confirms that the classical transition from the bubbles Phase IV to the monodomain Phase V is first-order in nature, since $\beta_0$ experiences a sudden jump, while the quantum-induced dynamical Phases IV′ and IV″ are bridging structures between Phases IV and V -- allowing a continuous variation of $\beta_0$ from Phase IV to Phase V.

## Quantum critical fluctuations

Generally, continuous phase transitions at low temperatures are associated with a quantum critical point (QCP)[40]. The latter pins a critical value of an external parameter at which quantum fluctuations, in analogy with their thermal counterpart, lead to a spontaneous symmetry breaking. Examples of QCPs include quantum phase transitions in Ising ferromagnet LiHoF$_4$[41], high T$_c$ cuprate superconductor La$_{1.86}$Sr$_{0.14}$CuO$_4$[42] and double layer Quantum Hall systems[43]. While the QCP occurs at absolute zero, its presence can be also detected at finite temperatures. As temperature increases, the QCP fans out into a finite range of external parameter values wherein quantum critical fluctuations play a dominant role[40]. Consequently, at finite low temperature, one has a quantum critical region, rather than a quantum critical point (see Supplementary Note 7).

The characteristic features of such quantum critical region can significantly vary with the dimensionality of the system and can even depend on the particularities of the microscopic interactions[44–46]. In

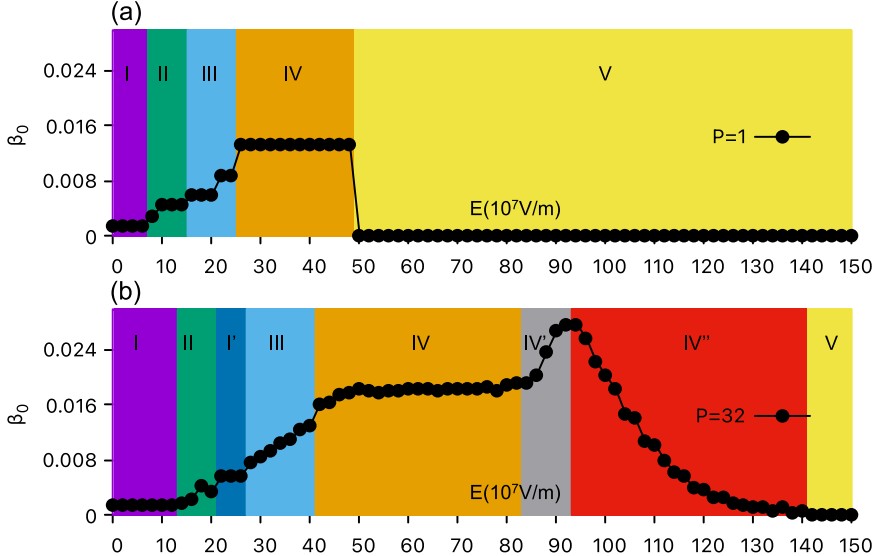

**Fig. 3 | The zeroth Betti number ($\beta_0$) as a function of electric fields. a** CMC ($P = 1$) simulations. **b** PI-QMC ($P = 32$) simulations.

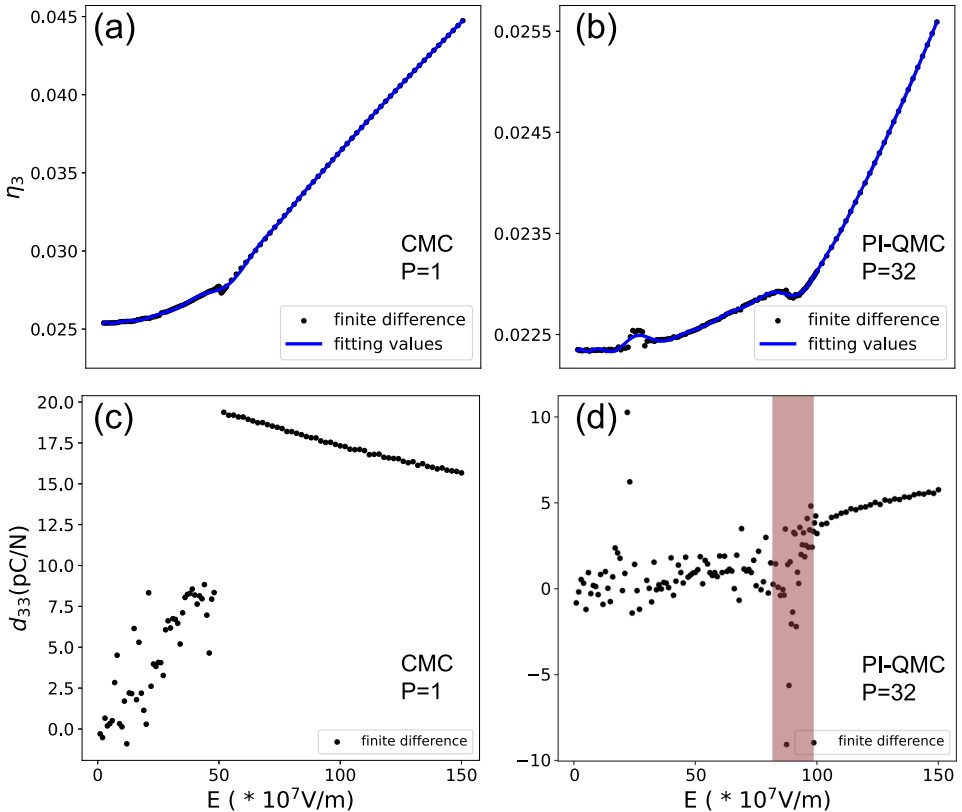

**Fig. 4 | Piezoelectricity from CMC and PI-QMC.** The $\eta_3$ strain as a function of electric field for CMC ($P=1$) (**a**) and PI-QMC ($P=32$) (**b**) simulations. The resulting piezoelectric coefficient ($d_{33}$) as a function of electric field for CMC (**c**) and PI-QMC (**d**) computations. The red region in (**d**) represents the quantum critical fluctuations region.

our case, a relevant and simple example is given by the one-[47] and two-dimensional Ising models in transverse field[48,49]. Both models feature a QCP corresponding to the quantum transition between the ordered ground state (ferromagnetic or antiferromagnetic) with spins oriented along the z-axis and a polarized state wherein the spins are oriented along the applied field direction (x- or y-axis). In this case, the quantum criticality leads to a universal scaling of the zero-momentum dynamical spin response function and scale invariance of the spatial spin correlations. Moreover, for these models a qualitative picture of critical fluctuations can be obtained by examining the crossover of spin excitations[50]. Specifically, at finite temperatures and away from the QCP, the excitations in the spin-ordered state correspond to dynamical domain walls while at high external fields the quasiparticles can be described as single spins oriented in the direction opposite to the field rather than domain walls. In contrast, within the quantum critical region, the scale invariance merges both scenarios – the critical spin dynamics can be visualized as an interacting gas of either isolated spins flipped against the applied field or moving domain walls.

In our case, the mere fact that the low-temperature transition between Phases IV and V is continuous and driven by quantum fluctuations already constitutes a strong evidence of a quantum critical behavior. We also find further qualitative considerations supporting quantum criticality by comparing our results with the quantum transition in the 2D Ising model. Particularly, the resemblance can be clearly seen in the observed character of dipolar excitations within the critical region - at the lower field boundary of Phase IV', the QFs-induced dynamics of individual dipoles are highly correlated and tied to the dynamics of polar bubbles or, equivalently, domain walls. At the same time, the fluctuations at the higher field end of Phase IV'' can be rather described as dynamical flips of standalone dipoles. Moreover, the crossover region hosting mixed excitations can be clearly identified from Fig. 3b as a region of increased $\beta_0$ corresponding to the field

values from $84 \times 10^7$ V/m to $100 \times 10^7$ V/m. In this region, the size of bubbles fluctuates significantly and can even decrease to a point where bubbles degenerate into single dipoles oriented against the external bias. Such scale invariance, as well as, the mixed dynamical excitations picture hints to the presence of quantum critical fluctuations for $84 \times 10^7$ V/m $< E_z < 100 \times 10^7$ V/m.

## Piezoelectricity from CMC and PI-QMC

Let us now investigate if some physical properties, such as piezoelectricity, are affected by ZPPVs. Figure 4a shows the strain ($\eta_3$) along the out-of-plane direction as a function of electric field for CMC at 20 K. The piezoelectric coefficient $d_{33}$ is then obtained as the derivative of $\eta_3$ with respect to that field and is shown in Fig. 4c. From $E_z = 0 \times 10^7$ V/m to $E_z = 48 \times 10^7$ V/m (that corresponds to the range of stabilities of labyrinthine, disconnected labyrinths, mixed bimerons-bubbles and bubbles, i.e., Phases I, II, III and IV), $\eta_3$ increases with increasing electric field yielding positive $d_{33}$. The piezoelectric coefficient of the bubble phase IV is about 8.0 pC/N. A first-order transformation then occurs near $E_z = 48 \times 10^7$ V/m, when the bubbles transform into monodomains. When the field is strengthened within the monodomain phase, the strain increases and $d_{33}$ is thus positive too (and larger than that of bubbles) – with this piezoelectric coefficient slightly and nearly linearly decreasing from 19.4 to 15.7 pC/N when E varies between $52 \times 10^7$ V/m to $150 \times 10^7$ V/m.

We now turn to the PI-QMC case, with $P = 32$. The corresponding $\eta_3$ and $d_{33}$ are shown as a function of electric field in Fig. 4b, d, respectively. One can first see that up to fields of $142 \times 10^7$ V/m, that is before reaching the monodomain phase, $\eta_3$ mostly only slightly depends on the electric field, implying that ZPPVs have the general tendency to reduce the piezoelectric response. However, two exceptions occur to that general trend within this range. One for $E_z = 22 \times 10^7$ V/m, which corresponds to the transition between

disconnected labyrinths and dynamic stripes (Phase I') for which $\eta_3$ suddenly increases with field and thus generates a positive peak in $d_{33}$. The second exception happens in the aforementioned quantum critical region for which $84 \times 10^7$ V/m $< E_z < 100 \times 10^7$ V/m. One can see there a strong dependency of $\eta_3$ and $d_{33}$ with field, with even a strong negative peak for $d_{33}$ near $E_z = 94 \times 10^7$ V/m. Having negative longitudinal piezoelectricity is a rather rare fundamental feature in piezoelectric materials (see, e.g., refs. [26,51–53]) and may have some applications[54]. Since both Phases I' and the quantum critical region only exist in the quantum phase diagram of Fig. 2b and not in the classical one depicted in Fig. 2a, one can conclude that quantum effects dramatically affect the piezoelectric response of the investigated system, by inducing strong positive and negative peaks at specific electric fields, respectively. When the electric field is larger than $100 \times 10^7$ V/m within PI-QMC simulations, the resulting dipolar liquid phase now adopts a $d_{33}$ that smoothly and slightly increases when further strengthening this field. A smooth field-induced variation of $d_{33}$ also occurs in the monodomain phase V when $P = 32$ in the PI-QMC computations. Note that we also calculated dielectric properties from CMC and PI-QMC simulations as a function of electric fields (Supplementary Note 8). One can see, in particular, a decrease of the dielectric susceptibility at the transition between disconnected labyrinths and dynamic stripes, and a strong dependency of such susceptibility, including a peak, in the quantum critical region.

## Discussion

We combined an atomistic effective Hamiltonian with PI-QMC to investigate the effects of ZPPVs on the various quenched topological patterns of ultrathin ferroelectric PZT films under electric field. Compared with the classical case, the QFs right shift and broaden the regions of the electric field stability of different topological patterns. This effect is particularly pronounced in the case of the bubble phase. Moreover, our simulations indicate that ZPPVs can result in strong dipolar fluctuations at low temperatures and thus entail new quantum phenomena. Namely, at lower bias field magnitudes, such fluctuations lead to the field-induced inverse transition from disconnected polar labyrinths to dynamic stripe-like state. Additionally, we show that strong QFs can induce two novel phases that can be described as bubble liquids and dipolar liquids. The properties of these two quantum-induced states are determined by ZPPVs which enable dynamics of either polar bubbles or standalone dipoles and allow for a continuous quantum transition between the polar bubble lattice and the homogeneously polarized monodomain pattern. Another striking result is the discovery of a quantum critical region involving the bubble liquids and dipolar liquids, with this region having an unusual behavior, and strong field-dependency, of some physical properties – including a negative longitudinal piezoelectricity. We are confident that these findings broaden our knowledge of quantum effects, topology, ferroelectricity and nanostructures, and hope that our predictions will be verified soon by experiments.

## Methods

We mimic ferroelectric ultrathin films made of PZT, that are grown along the [001] pseudo cubic direction (which is chosen to be the z-axis). All surfaces and interfaces are taken to be Pb-O terminated. These PZT films have a thickness of about 2 nm (that is, 5 unit cells) and are under a compressive strain of about −2.65%, to mimic the growth on a $SrTiO_3$ substrate. The simulation supercell is $26 \times 26 \times 5$ and is periodic along the [100] and [010] pseudo-cubic in-plane directions, while finite along the z-axis. The electrical boundary condition is set to screen 80% of the polarization-induced surface charges, which is realistic. The effect of the electric field on physical properties is included by adding an energy term that is minus the dot product between the electrical polarization and the external electric

field, $E_z$. This latter is applied along the z-axis and ranges from $0 \times 10^7$ V/m (no field) to $150 \times 10^7$ V/m in magnitude in our simulations. Note that effective Hamiltonian schemes usually have the tendency of providing a higher field than experiments by one order of magnitude[55,56], which is likely related to Landauer paradox[57]. This may thus be due to inhomogeneities in experiments, while the calculations consider a defect-free medium. Note that the piezoelectric response or any other physical response can be affected by the presence of structural defects, as, e.g., shown in ref. [58]. The total number Monte Carlo sweeps (MCS) is chosen to be 200,000 with thermalization being accomplished over the first 150,000 sweeps and averaging of physical properties being made over the last 50,000 sweeps.

Technically, the first-principle-based effective Hamiltonian of Refs. [8,29,59] is used within CMC and PI-QMC simulations to determine energetics and configurations. Previous CMC simulations based on such effective Hamiltonian have predicted that 180° nanostripe domains[29,60] can be the ground state for PZT films under compressive strain and open-circuit-like boundary conditions, which is consistent with, e.g., the experimental results of Ref. [61]. Other CMC simulations using such effective Hamiltonian further predicted many electrical topological defects, including vortices[8], merons[19], dipolar waves[16] and bubbles/skyrmions[29], that were then experimentally confirmed[9,10,17,19].

To consider the ZPPVs on electrical topological patterns, the PI-QMC approach is used. In this approach, each quantum particle (associated with five-atom cell) interacts with its own images at neighboring imaginary time slices through a spring-like potential, while all quantum particles interact with each other at the same imaginary time slice via the effective Hamiltonian. The number of imaginary times is called the Trotter number, is denoted as $P$ here and takes the maximum value of 32 in our present work, in order to have convergent results at the (low) temperature T of 20 K[4,62,63]. The Trotter number is defined by the Trotter product formula[64,65] for two non-commuting operators $\hat{A}$ and $\hat{B}$: $\exp\left(\hat{A} + \hat{B}\right) \underset{P \to \infty}{\to} \left[\exp(\hat{A}/P)\exp(\hat{B}/P)\right]^P$, where $P$ is the trotter number. Specifically, for a single particle moving in a potential $V$, the Trotter formula is $\exp[-(\hat{E}_{kln} + \hat{V})/k_B T] = \lim_{P \to \infty} \left[\exp\left(-\hat{E}_{kln}/k_B TP\right)\exp(-\hat{V}/k_B TP)\right]^P$. $\hat{E}_{kln}$ and $\hat{V}$ are kinetic and potential operators for the particle.

## Data availability

The authors declare that the coefficient data of the effective Hamiltonian are available within the paper and its Supplementary material. The data that support the findings of this study are available from the corresponding author upon request.

## Code availability

The code used in this study is available from the corresponding authors upon request.

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

## Acknowledgements

This project was supported by the Vannevar Bush Faculty Fellowship (VBFF) Grant No. N00014-20-1-2834 from the Department of Defense, and we are also thankful for the support from the MonArk Quantum Foundry supported by the National Science Foundation Q-AMASE-i program under NSF award no. DMR-1906383. We also acknowledge the computational support from the Arkansas High Performance Computing Center for computational resources.

## Author contributions

L.B. and S.P. conceived the concept and designed the project. W.L. performed the atomic simulations with the help from A.A., S.P. and Y.N. W.L. prepared the initial draft of the paper. S.P. and Y.N. wrote the quantum criticality part. All authors contributed to the writing and revision of the paper.

## Competing interests

The authors declare no competing interests.
