## [Peer Review File · Nature Communications]

REVIEWER COMMENTS

Reviewer #1 (Remarks to the Author):

In the article "Quantum criticality at cryogenic melting of polar bubble lattices" the authors explore the phase diagram of an effective Hamiltonian for ultrathin PZT films and compare Classical and Quantum Monte Carlo simulations. They report different ferroelectric domain patterns predicted by the model and how the zero-point energy vibrations affect the phase diagram. In particular, they show that this quantum effect shifts and enhances the stability regions of the phases, and is even responsible for the appearance of new phases. They also compute the piezoelectric response as a function of field, finding negative longitudinal piezoresponse in some regimes.

The work is original and scientifically sound. I believe the article is well written and it improves the understanding of domain textures in ferroelectric thin films. However, I also believe some points should be clarified before publication.

1. Because of the delicate balance of the different energy terms, it would be important to understand the role of the simulation box size in the results. Have the authors checked how finite size effects affect their results? I am concerned in particular about the lateral size, since some domain patterns (like the stripes) seem to only accommodate to certain simulation box sizes.
2. The final temperature after quenching is low (20 K), but it is not clear if it is low enough to claim the existence of quantum critical points. Have the authors checked (even if only for a few selected cases) the effect of the final temperature on their results?
3. The quenching of the system is mentioned, but no details about the quenching/annealing rate are given. Have the authors checked how the annealing rate affects the resulting state? Also, I believe the authors should specify how fast the quenching is performed, so that reproducibility of their results is possible.
4. A bimeron is a topologically non-trivial excitation of a trivial (homogeneous) field that requires the dipoles in the periphery to lie in the plane, and a core with out-of-plane dipoles with a certain bimodal distribution. However, because the preferred direction of the polarization is out-of-plane (given the applied compressive strain and field) I believe what the authors are seeing in phase III has no meronic character, since the underlying homogeneous field and the local distortion are both pointing out-of-plane. Can the authors clarify what they mean by bimeron and how they verified

these domains have a bimeronic character? If they confirm the bimeronic nature of the domains, the authors should show the in-plane component of the dipoles. If not, I recommend renaming phase III. Note that I believe the use of "meron" in topological ferroelectric patterns has been seldom misused in the literature by other authors.

5. The Betti number takes continuous values between 0 and 0.030, and not discrete steps. This is particularly true when ZPPVs are considered (Fig. 3b). Topological invariants, on the contrary, take only discrete values (typically integers). Therefore, I believe in this case the Betti number cannot be deemed a topological invariant, even if it is useful when classifying the domain patterns.

6. What is the difference between the labyrinths and the disconnected labyrinths phases? From the figures they seem the same phase (for example in Fig.2a1 there is a bright "L"-shaped domain in the centre that is disconnected, essentially identical to the "I"-shaped dark disconnected domain in Fig.2a2). The Betti number of both phases seems different, but this quantity is essentially continuous, and therefore the distinction is not clear to me.

7. The fit in Fig. 4 conveys the message of having found a physical model that describes the evolution of the strain and piezoresponse with electric field. However, fitting to polynomials of order 60 (Fig. 4) is almost certainly going to lead to overfitting. Most importantly, it provides very little physical insight (specially when the fitting is guided to capture certain points, as done in this work). I therefore suggest finding a valid (physical) reason for the fit or refrain from using it.

8. In the introduction, the authors state that "Answering these questions [...] may also lead to novel technologies.", but no particular technology is mentioned. Could the authors give an example of what kind of technologies they have in mind?

9. The Trotter number, and to a certain extent the Betti number, is used in the text with very little introduction. I believe these quantities are not common knowledge even among specialized readers. In order to make the text more accessible, I suggest briefly introducing them.

10. The authors state that "The criterion for convergence is, in fact, that the Betti number (to be discussed later) is merely unaffected by P." However, I have not found the discussion of this point later in the text. Can the authors clarify this point?

Small typos I found:

- i. “the discontinuous classical behavior of [...] become continuous” -> “the discontinuous classical behavior of [...] becomes continuous”
- ii. “free -landscape” -> “free-energy landscape”
- iii. “have nearly vanished β_0 ” -> “have nearly vanishing β_0 ”

Reviewer #2 (Remarks to the Author):

In this manuscript, Bellaiche et al. investigated the Quantum fluctuations (QFs) caused by zero-point phonon vibrations for polar skyrmions in a PZT film. They discovered that the ZPPVs do not suppress polar patterns but rather stabilize the topological structures. This is a very interesting study, and I would recommend it for publication after major revisions. My comments:

1) Regarding the calculations of the piezoelectric coefficient. Typically, experimentally, the piezoelectric coefficient is measured after poling with a medium field and relax so it won't be zero (or close to zero). I would wonder if the authors can calculate the butterfly loop to quantify the piezoelectric coefficient. The negative piezoelectric coefficient (although we see them in PVDF) might be an artifact.

2) Regarding the degree of screening. Here 80% screening is assumed. Experimentally, the degree of screening depends on the growth condition and whether it is exposed to air. It would be better if the authors can discuss the influence of this parameter as well.

3) The electric field in this study is way larger than the breakdown field (Typically ~ 3000 kV/cm). And the actual threshold coercive field is ~ 1500 kV/cm (Nature Materials, 20,194–201, 2021). Would be valuable if the authors can discuss the factors that can influence the coercive field calculated by QMC, e.g., the presence of defects, degree of screening, temperature.

4) Some relevant literatures should be discussed and cited, for instance, the switching from Labyrinth to bubble has been studied in Acta Mater. 152, 155-161 (2018); Matter, 5, 3, 1031-1041, 2022

and the possible topological transitions have been reviewed in Small Methods, 2200486 (2022)

Rebuttal letter to referees (NCOMMS-23-22122-A)

We would like to thank the Referees for their helpful questions and comments. In the process of answering them and revising the text, our manuscript has been improved. Below, we address all their concerns in detail. The resulting modifications to the manuscript clarify some of the most important points of our work. The revisions and additions made in the manuscript are highlighted by red colored font in the marked-up version.

For each Referee, we reply to each question raised by first quoting the question followed by our response.

Response to Referee 1

Comment 1: In the article "Quantum criticality at cryogenic melting of polar bubble lattices" the authors explore the phase diagram of an effective Hamiltonian for ultrathin PZT films and compare Classical and Quantum Monte Carlo simulations. They report different ferroelectric domain patterns predicted by the model and how the zero-point energy vibrations affect the phase diagram. In particular, they show that this quantum effect shifts and enhances the stability regions of the phases, and is even responsible for the appearance of new phases. They also compute the piezoelectric response as a function of field, finding negative longitudinal piezoresponse in some regimes.

The work is original and scientifically sound. I believe the article is well written and it improves the understanding of domain textures in ferroelectric thin films. However, I also believe some points should be clarified before publication.

Reply 1: We thank this Referee for mentioning that "The work is original and scientifically sound. I believe the article is well written and it improves the understanding of domain textures in ferroelectric thin films". Below the questions and comments posed by the Referee are fully addressed.

Comment 2: Because of the delicate balance of the different energy terms, it would be important to understand the role of the simulation box size in the results. Have the authors checked how finite size effects affect their results? I am concerned in particular about the lateral size, since some domain patterns (like the stripes) seem to only accommodate to certain simulation box sizes.

Reply 2: We thank the Referee for this constructive suggestion. We have performed the calculations for a serial of lateral sizes, *i.e.*, $24 \times 24 \times 5$, $25 \times 25 \times 5$, $26 \times 26 \times 5$, $28 \times 28 \times 5$, $30 \times 30 \times 5$ and $32 \times 32 \times 5$. For CMC and PI-QMC, their energy differences are listed in Table 1. Specifically, we thermalize the system at 500 K, and then quench the system to 20 K. The final configurations are all labyrinths. From the Table, one can

see that the energy differences between different lateral sizes are very small for both CMC and PI-QMC ($\sim 1\text{meV}$ per atom), indicating that the size effects are not very important here (since the initial simulation box was large enough).

Table 1: Energy differences for different lateral sizes from CMC and PI-QMC simulations. In our work, we use the 26×26 lateral size and set its energy to be zero.

	24×24	25×25	26×26	28×28	30×30	32×32	unit
CMC	0.9	0.8	0	1.2	0.9	0.9	meV/atom
PI-QMC	0.5	0.5	0	-0.1	0	0.3	meV/atom

Comment 3: The final temperature after quenching is low (20 K), but it is not clear if it is low enough to claim the existence of quantum critical points. Have the authors checked (even if only for a few selected cases) the effect of the final temperature on their results?

Reply 3: Following the Referee’s suggestion, we conducted calculations at 40K, 35K, 30K, 25K and 15K (Note that it is difficult to do simulations at lower temperatures since it will require even larger trotter number, and thus time-consuming calculations, to converge the results). In fact, the results for these temperatures of 40K, 35K, 30K, 25K and 15K already give the same qualitative results: a quantum critical region does locate in-between phase IV’ (bubble liquids) and phase IV’’ (dipolar liquids). Let us also emphasize that, for a finite temperature, the quantum critical point at 0K evolves to be a quantum critical region that grows in size as the temperature increases (see below in Figure 1a). In our case, the quantum critical region can be identified as a region involving Phase IV’ and IV’’ (that range in-between phase IV, for which bubbles whose zeroth Betti number almost do not change as the electric field increases, and the monodomain of Phase V. More precisely, we plot in Fig. 1b the zeroth Betti number as a function of electric field and the black and red double-headed arrows provide a measure of the quantum critical region, for $T=15\text{K}$ and $T=40\text{K}$ cases, respectively. One can indeed see that $T=40\text{K}$ has a wider quantum critical region than $T=15\text{K}$. From Figure 1b, one can further see that the quantum critical region continuously shrinks as the temperature decreases from 40K to 15K, as consistent with Fig. 1a. Thus, we can expect that the quantum critical region will become a quantum critical point as the temperature reaches to 0K. However, simulating the 0K properties is not computationally feasible.

Figure 1: (a) Schematic diagram of a quantum phase transition. In our case, the x axis represents the direct current electric field along the z -direction. (b) The zeroth Betti number as a function of electric fields. Different colors represent different temperature cases. Here, the trotter number is $P=32$. The black and red double-headed arrows provide a measure of the quantum critical region for $T=15\text{K}$ and $T=40\text{K}$ cases.

Comment 4: The quenching of the system is mentioned, but no details about the quenching/annealing rate are given. Have the authors checked how the annealing rate affects the resulting state? Also, I believe the authors should specify how fast the quenching is performed, so that reproducibility of their results is possible.

Reply 4: Yes, the results indeed depend on the cooling rate. There are two limit cases. The one is the instantaneous quench (extremely very fast cooling) which we used in this work. Specifically, we first thermalize the system at 500K , and then quench to 20K . The fastest dipole relaxation time is about $\tau=50\text{fs}$ (npj Computational Materials 4, 80, (2018)). Hence, in our simulations, the quench rate is roughly $R_{\text{fast}} = \Delta T / \tau \sim 500\text{K}/50\text{fs} = 10\text{K/fs}$ or faster. In this case, we get the labyrinths by quenching the system from 500K to 20K .

The other limiting case is the slow cooling (annealing). It would roughly correspond to $R_{\text{slow}} = 5\text{K}/0.5\text{ns} = 10\text{K/ns}$. In this case, we get the stripe by annealing from 500K to 20K (the temperature step is 5K). Cooling rate much below 10K/ns are not accessible in neither MD nor MC simulations due to large calculations.

Comment 5: A bimeron is a topologically non-trivial excitation of a trivial (homogeneous) field that requires the dipoles in the periphery to lie in the plane, and a core with out-of-plane dipoles with a certain bimodal distribution. However, because the preferred direction of the polarization is out-of-plane (given the applied compressive strain and field) I believe what the authors are seeing in phase III has no meronic character, since the underlying homogeneous field and the local distortion are both pointing out-of-plane. Can the authors clarify what they mean by bimeron and how they verified these domains have a bimeronic character? If they confirm the bimeronic nature of the domains, the authors should show the in-plane component of the dipoles.

If not, I recommend renaming phase III. Note that I believe the use of "meron" in topological ferroelectric patterns has been seldom misused in the literature by other authors.

Reply 5: This is a good point too. By bi-merons, we mean structures like the ones reported in spin systems in Ref. [Phys. Rev. B 83, 100408(R), (2011)]. We have also verified that, in our case, each meron carries a fractional Skyrmion charge of $1/2$ [Nature communications, 11, 5779 (2020) and Review of Modern Physics, 95, 025001, (2023)]. Albeit we only show the distribution of the out-of-plane components, there are also in-plane dipole components at the “domain walls”. In fact, at the stripe, meron, or bubble boundaries, the dipoles rotate in a Néel fashion. To show this, we plot the in-plane and out-of-plane components of the 4th plane of the film (under an electric field $E=24\times 10^7\text{V/m}$), which is shown in Figure 2b below. The magnitude of the arrows represents the in-plane components. The color characterizes the magnitude of the out-of-plane components. The blue dashed and red dashed regions represent the bi-merons and bubble, respectively. For bi-merons, the dipoles in the periphery indeed lie in the plane and possess a core with out-of-plane dipoles with a certain bimodal distribution. Figure 2a is similar to Figure 2b, but just show the out-of-plane components.

Figure 2 (a) The out-of-plane components of local modes u_z (in atomic units) for the 4th plane of the PZT film (under electric field $E=24\times 10^7\text{V/m}$ with $P=1$). (b) The in-plane and out-of-plane components of local modes u_z (in atomic units) for the 4th plane of the PZT film (under electric field $E=24\times 10^7\text{V/m}$ with $P=1$). The magnitude of the arrows represents the in-plane components. The color means the magnitude of the out-of-plane components. The blue dash and red dash regions represent the bi-merons and bubble. For bi-merons, the dipoles in the periphery lie in the plane, and a core with out-of-plane dipoles with a certain bimodal distribution.

Comment 6: The Betti number takes continuous values between 0 and 0.030, and not discrete steps. This is particularly true when ZPPVs are considered (Fig. 3b). Topological invariants, on the contrary, take only discrete values (typically integers).

Therefore, I believe in this case the Betti number cannot be deemed a topological invariant, even if it is useful when classifying the domain patterns.

Reply 6: This is a good question as well. By definition, Betti numbers always take integer values. For example, the zero-th Betti number corresponds to the number of connected components of the manifold. In our case, as the manifold we take the set of points where the dipoles are oriented against the applied bias. As a result, the zero-th Betti number simply becomes the number of domains oriented against the bias. For possible comparison with experiment, we plotted the Betti numbers divided by the area of the supercell (that is, the density of the Betti numbers) which is now a non-integer. This is analogous to using, *e.g.*, the non-integer Skyrmion charge density instead of the integer Skyrmion charge.

Comment 7: What is the difference between the labyrinths and the disconnected labyrinths phases? From the figures they seem the same phase (for example in Fig.2a1 there is a bright "L"-shaped domain in the centre that is disconnected, essentially identical to the "I"-shaped dark disconnected domain in Fig.2a2). The Betti number of both phases seems different, but this quantity is essentially continuous, and therefore the distinction is not clear to me.

Reply 7: The detailed discussion of such differences can be found in [Nature communications, 11, 5779 (2020)] and its supplemental information. For disconnected labyrinths, the probability of observing a percolating domain polarized against the bias becomes equal to zero. In relation with this, at the transition between labyrinths and disconnected labyrinths, the fractal dimension of the domain set polarized against the bias coincides with the fractal dimension of self-avoiding random walks in 2D. Also, at the transition between labyrinths and disconnected labyrinths, the density of the zeroth Betti number features an inflection point.

Comment 8: The fit in Fig. 4 conveys the message of having found a physical model that describes the evolution of the strain and piezoresponse with electric field. However, fitting to polynomials of order 60 (Fig. 4) is almost certainly going to lead to overfitting. Most importantly, it provides very little physical insight (specially when the fitting is guided to capture certain points, as done in this work). I therefore suggest finding a valid (physical) reason for the fit or refrain from using it.

Reply 8: Near transition points, d_{33} fluctuates a lot (since it is a derivative of η_3) and is very sensitive to electric field. Hence, it is not easy to fit the curve. Following the Referee's suggestion, we now use the raw data for d_{33} in Figure 4c(CMC) and 4d (PI-QMC) of the main text (and also the raw data for χ_{33} in Figure S8c (CMC) and S8d (PI-QMC) in supporting information).

Comment 9: In the introduction, the authors state that "Answering these questions [...] may also lead to novel technologies.", but no particular technology is mentioned. Could

the authors give an example of what kind of technologies they have in mind?

Reply 9: We thank the Referee’s constructive suggestions. Now, we have added “For instance, the discovery of correlated quantum phases in low-dimensional polar systems could enable new opportunities for quantum³¹ and neuromorphic computing³²” and highlight them with red color.

Comment 10: The Trotter number, and to a certain extent the Betti number, is used in the text with very little introduction. I believe these quantities are not common knowledge even among specialized readers. In order to make the text more accessible, I suggest briefly introducing them.

Reply 10: We are thankful for this Referee’s suggestion too. We have now added “The Trotter number is defined by the Trotter product formula^{64,65} for two non-commuting operators \hat{A} and \hat{B} : $\exp(\hat{A} + \hat{B}) \xrightarrow{P \rightarrow \infty} [\exp(\hat{A}/P)\exp(\hat{B}/P)]^P$, where P is the trotter number. Specifically, for a single particle moving in a potential V , the Trotter formula is $\exp[-(\widehat{E}_{kin} + \widehat{V})/k_B T] = \lim_{P \rightarrow \infty} \{\exp(-\widehat{E}_{kin}/k_B TP) \exp(-\widehat{V}/k_B TP)\}^P$.

\widehat{E}_{kin} and \widehat{V} are kinetic and potential operators for the particle” in the main text. We also added “In algebraic topology, the Betti numbers are used to distinguish topological spaces based on the connectivity of n-dimensional simplicial complexes[3]. The k^{th} Betti number refers to the number of k -dimensional holes on a topological surface. A “ k -dimensional hole” is a k -dimensional cycle that is not a boundary of a $(k+1)$ -dimensional object. The first few Betti numbers have the following definitions for 0-dimensional, 1-dimensional, and 2-dimensional simplicial complexes: β_0 is the number of connected components (which we used in our work); β_1 is the number of one-dimensional or “circular” holes; β_2 is the number of two-dimensional “voids” or “cavities”” and highlight them with red color.

Comment 11: The authors state that “The criterion for convergence is, in fact, that the Betti number (to be discussed later) is merely unaffected by P.” However, I have not found the discussion of this point later in the text. Can the authors clarify this point?

Reply 11: This point is shown in Figure S5 of the Supporting information. Now, we also indicated it in main text by adding “One also can see the convergence trend as a function of Trotter number” in paragraph 1 of page 7 in the main text, and highlight this addition with red color.

Comment 12: Small typos I found

- i. “the discontinuous classical behavior of [...] become continuous” -> “the discontinuous classical behavior of [...] becomes continuous”
- ii. “free -landscape” -> “free-energy landscape”

iii. “have nearly vanished β_0 ” -> “have nearly vanishing β_0 ”

Reply 12: We appreciate the Referee’s carefully reading. In the new version, we have corrected these typos.

Response to Referee 2

Comment 1: In this manuscript, Bellaiche et al. investigated the Quantum fluctuations (QFs) caused by zero-point phonon vibrations for polar skyrmions in a PZT film. They discovered that the ZPPVs do not suppress polar patterns but rather stabilize the topological structures. This is a very interesting study, and I would recommend it for publication after major revisions.

Reply 1: We thank the Referee for mentioning that “This is a very interesting study, and I would recommend it for publication after major revisions”. Below the questions and comments posed by this Referee are fully addressed.

Comment 2: Regarding the calculations of the piezoelectric coefficient. Typically, experimentally, the piezoelectric coefficient is measured after poling with a medium field and relax so it won't be zero (or close to zero). I would wonder if the authors can calculate the butterfly loop to quantify the piezoelectric coefficient. The negative piezoelectric coefficient (although we see them in PVDF) might be an artifact.

Reply 2: We thank the Referee for this constructive suggestion. We calculate the butterfly loop with trotter number $P=32$ [note $P=1$ (classical simulations) does not yield a negative longitudinal piezoelectric effect. The negative longitudinal piezoelectric effect is induced by quantum fluctuation] near the negative piezoelectric region. The results are shown in Figure 3 (See below, note that the calculations for PI-QMC with $P=32$ are time-consuming. Thus, we have sparse points). From it, one can clearly see prominent negative slopes in the region ranging from about 84 to $86 \cdot 10^7$ V/m when increasing the electric field and in the region varying between about 86 and $85 \cdot 10^7$ V/m when decreasing this field. This data confirms a negative piezoelectric coefficient. Note that such piezoelectric coefficient has been calculated in Fig. 4d of main text, and its value is about -9.1 pC/N .

Figure 3: Butterfly loop for strain versus electric field, within the PI-QMC with P=32.

Comment 3: Regarding the degree of screening. Here 80% screening is assumed. Experimentally, the degree of screening depends on the growth condition and whether it is exposed to air. It would be better if the authors can discuss the influence of this parameter as well.

Reply 3: We thank the Referee for this suggestion. Indeed, topological patterns can be related with the screening factor. There are two extreme cases. The first is the ideal open-circuit (OC) condition, for which unscreened polarization-induced surface charges can generate a large depolarizing electric field along the growth direction. The second one is the ideal short-circuit (SC) condition, which is associated with a vanishing internal field resulting from the full screening of surface charges. The OC and SC conditions are characterized by $\beta_{oc}(\beta_{oc}=0)$ and $\beta_{sc}(\beta_{sc} = 1)$ in our methodology. Previous works (see, *e.g.*, [Physical Review Letters 93, 196104 (2004)]) indicate that below a certain value of β close to 85% for PZT film under compressive (-2.65%) strain, the results become independent of β (since the initial depolarizing field is large enough to not affect anymore properties, as a result of the fact that dipoles have adopted striking patterns to “counter-attack” it). This is confirmed by the calculations shown below in Fig. 3, in that the results of $\beta=80\%$ (which is the β used in the present work) are similar to those of $\beta=70\%$ and 60% , both for CMC (P=1), that gives labyrinths pattern, and PI-QMC (P=32), that also yields labyrinths but with smaller dipoles. On the other hand, a larger β of 90% makes the labyrinths pattern disappear in favor of monodomains for CMC (and dipolar liquid for PI-QMC), as consistent with short-circuit-like conditions. In that regime of larger β , the magnitude of the dipoles significantly depends on β since the residual depolarizing field is rather small within the monodomain but strongly dependent on this screening parameter.

Figure 3 (a) The topological patterns for different screening factor from CMC ($P=1$) and PI-QMC ($P=32$). The color bar indicates the z -component of the local modes u_z (in atomic units).

Comment 4: The electric field in this study is way larger than the breakdown field (Typically ~ 3000 kV/cm). And the actual threshold coercive field is ~ 1500 kV/cm (Nature Materials, 20,194–201, 2021). Would be valuable if the authors can discuss the factors that can influence the coercive field calculated by QMC, e.g., the presence of defects, degree of screening, temperature.

Reply 4: In experiments, there are usually inhomogeneities, such as defects, that affect the value of the coercive field. However, for our calculations, we consider a perfectly defect-free medium. As a result and based on the Landauer paradox [R. Landauer, J. Appl. Phys. 28, 227 (1957)], the electric field used in calculations can be up to 20 times larger than in experiments [please see Phys. Rev. Lett 103, 047204 (2009), Nat. Commun. 8, 15682 (2017), Phys. Rev. B 107, 214105 (2023)]. We have now indicated this in paragraph 1 of page 15 in the main text and highlight this explanation with red color.

Comment 5: Some relevant literatures should be discussed and cited, for instance, the switching from Labyrinth to bubble has been studied in Acta Mater. 152, 155-161 (2018); Matter, 5, 3, 1031-1041, 2022 and the possible topological transitions have been reviewed in Small Methods, 2200486 (2022)

Reply 5: We thank the Referee for emphasizing these excellent works. We are now citing them in the sentence “Electric field induced topological phase transition were also reported in other ferroelectric superlattices from classical simulations³⁵⁻³⁷” in paragraph 2 of page 4 in the main text and highlight such addition with red color.

The followings are the revisions we made in the main text and supporting information

1, For the response of comment 3 of Referee 1, we have added “*To verify the quantum critical behavior, we conducted calculations at 40K, 35K, 30K, 25K and 15K (Note we cannot do lower temperature case since it will require even larger trotter number, and thus time-consuming calculations, to converge the results). In fact, the results for these temperatures of 40K, 35K, 30K, 25K and 15K already give the same qualitative results: a quantum critical region does locate in-between phase IV’ (bubble liquids) and phase IV’’ (dipolar liquids). Let us also emphasize that, for a finite temperature, the quantum critical point at 0K evolves to be a quantum critical region that grows in size as the temperature increases (see Fig. S7a). In our case, the quantum critical region can be identified as a region involving Phase IV’ and IV’’ (that range in-between phase IV, for which bubbles whose zeroth Betti number almost do not change as the electric field increases, and the monodomain of Phase V). More precisely, we plot in Fig. S7b the zeroth Betti number as a function of electric field and the black and red double-headed arrows provides a measure of the quantum critical region, for $T=15K$ and $T=40K$ cases, respectively. One can indeed see that $T=40K$ has a wider quantum critical region than $T=15K$. From Fig. S7b, one can further see that the quantum critical region continuously shrinks as the temperature decreases from 40K to 15K, as consistent with Fig. S7a. Thus, we can expect that the quantum critical region will become a quantum critical point as the temperature reaches to 0K. However, simulating the 0K properties, is not feasible.*” in part VI of supporting information.

2, For the response of comment 4 of Referee 1, we added “*The quench rate is roughly 10 K/fs or faster*” in paragraph 2 of page 3 of main text.

3, For the response of comment 5 of Referee 1, we added “*bimerons-bubbles can be clearly seen from Fig. S2 of SI*” in paragraph 2 of page 4 of main text. We also added the following figures as figure S2 in supporting information.

4, For the response of comment 8 of Referee 1, we have updated Figures 4c and 4d in the main text. We also updated Figures S8c and S8d in the supporting information.

5, For the response of comment 9 of Referee 1, we have added “*For instance, the discovery of correlated quantum phases in low-dimensional polar systems could enable new opportunities for quantum³¹ and neuromorphic computing³²*” in paragraph 3 of page 2 in the main text.

6, For the response of comment 10 of Referee 1, we have added “*The Trotter number is defined by the Trotter product formula^{64,65} for two non-commuting operators \hat{A} and \hat{B} : $\exp(\hat{A} + \hat{B}) \xrightarrow{P \rightarrow \infty} [\exp(\hat{A}/P)\exp(\hat{B}/P)]^P$, where P is the trotter number.*

Specifically, for a single particle moving in a potential V , the Trotter formula is $\exp[-(\widehat{E}_{kin} + \widehat{V})/k_B T] = \lim_{P \rightarrow \infty} \{ \exp(-\widehat{E}_{kin}/k_B T P) \exp(-\widehat{V}/k_B T P) \}^P$. \widehat{E}_{kin} and

*\widehat{V} are kinetic and potential operators for the particle” in the paragraph 3 of page 15 in the main text. We also added “*In algebraic topology, the Betti numbers are used to distinguish topological spaces based on the connectivity of n -dimensional simplicial complexes[3]. The k^{th} Betti number refers to the number of k -dimensional holes on a topological surface. A “ k -dimensional hole” is a k -dimensional cycle that is not a boundary of a $(k+1)$ -dimensional object. The first few Betti numbers have the following definitions for 0-dimensional, 1-dimensional, and 2-dimensional simplicial complexes: β_0 is the number of connected components (which we used in our work); β_1 is the number of one-dimensional or “circular” holes; β_2 is the number of two-dimensional “voids” or “cavities” in part IV of the supporting information.**

7, For the response of comment 11 of Referee 1, We have added “*One also can see the convergence trend as a function of Trotter number*” in paragraph 1 of page 7 in the main text.

8, For the response of comment 3 of Referee 2, We have added “*We also discuss the effects of different screening factors, see part I and Fig. S1 of supporting information*

(SI)” in paragraph 2 of page 3 in the main text. We also added the following figures as figure S1 in supporting information.

9, For the response of comment 4 of Referee 2, we have added “*Note that effective Hamiltonian schemes usually have the tendency of providing a higher field than experiments by one order of magnitude^{56,57}, which is likely related to Landauer paradox⁵⁸. This may thus be due to inhomogeneities in experiments, while the calculations consider a defect-free medium*” in paragraph 1 of page 15 in the main text.

10, For the response of comment 5 of referee 2, we have added “*Electric field induced topological phase transition were also reported in other ferroelectric superlattices from classical simulations³⁵⁻³⁷*” in paragraph 2 of page 4 in the main text.

REVIEWER COMMENTS

Reviewer #1 (Remarks to the Author):

The authors have successfully and carefully addressed most of the points I raised. However, I still believe two issues should be clarified before publishing:

1. After reading the authors' response, my main concern is that the authors claim that the results depend on the cooling rate employed, and that they have obtained the phase diagrams in this study using an extremely fast cooling.

Rapid quenching typically results in the system getting trapped in local minima of the free energy, which most of the times does not correspond to the experimental situation (in which the system shows the structure corresponding to the *global* minimum of the free energy). If the phases described in the manuscript are not the actual global minima of the system at a given temperature, then the relevance of this work would be much narrower, since it would be describing metastable phases of ferroelectric thin films (possibly experimentally accessible via extremely fast cooling), but not truly stable states. In fact, in their response the authors mention that for slower cooling rates they obtained a different phase, stripes instead of labyrinths. A similar result was reported by some of the authors in Nature 577, 47, where some of the phases in the simulation were only obtained under very fast annealing.

Could the authors elaborate on the justification for using an extremely fast cooling rate? In particular, can the authors compare the free energies of the labyrinth domain obtained with the very fast quenching (results of the main text) and the stripe domain state obtained with the slower quenching (mentioned in the report) at the same (final) temperature? I assume, although it is not completely clear to me from the authors' response in Reply 4, that this comparison is at zero field. Then, if the free energy of the labyrinth is larger than that of the stripes at zero field, that would mean that the authors' model predicts the more ordered phase (stripes) to be more stable than the more disordered phase (labyrinth), painting a very different picture from that described in the present work, which would deserve to be mentioned in the text.

2. A smaller concern of mine is that of the bimeronic character of phase III and of the definition of the bimeron itself. In the references pointed out by the authors in their response (Reply 5), bimeronic topological structures are depicted and described. Still, in one of the references [Nature

Communications, 11, 5779] some of the authors are also authors of the present manuscript and deal with essentially the same structure as in this work - I assume the reasoning behind calling these objects bimerons has probably been inherited from that work. In another reference [Review of Modern Physics, 95, 025001] that previous work is cited and partly reproduced, but no additional description or insight on the bimeronic structure of this phase is given. In the other referenced article in the response [Phys. Rev. B 83, 100408(R)], the structure depicted is homeomorphic to a (Bloch) skyrmion.

In other articles bimerons are described as objects with an in-plane periphery and a bimodal out-of-plane core:

- Phys. Rev. B 99, 060407(R) (2019)
- npj Comput Mater 6, 169 (2020)
- Appl. Phys. Lett. 118, 052411 (2021)
- Opt. Lett. 46, 3737-3740 (2021)
- Physics Reports, 895, 1 (2021)

In all these reports bimerons are shown as a bimodal distortion at the core perpendicular to its periphery (where the periphery should be extensible to the infinity). However, this does not correspond to the object that the authors describe in Fig. S2 (blue dashed line); instead the structure in the present work resembles a Néel-type skyrmion rather than a bimeron.

Then, and in order to clarify this issue, I would request from the authors to show the computed topological charge (with an explicit definition of the computed quantity) from which they deduce that the object enclosed by the blue dashed lines in Fig. S2 is a bimeron. (Note also that the skyrmionic number of a bimeron may be equal to 1, see e.g. Physics Reports, 895, 1, so computing the skyrmionic number may not be sufficient to distinguish between a bimeron and a skyrmion).

Otherwise, the findings of the article are still valid - but I would then insist on using a different name for the objects found in phase III.

3. List of small typos:

Line 48: "quantitively" -> "quantitatively"

Line 114: the reference to Fig. S1 should be to Fig. S2 instead.

Reviewer #2 (Remarks to the Author):

The authors have addressed all my comments, I'm happy to recommend for publication. Two more comment: (1) Would be valuable if the authors can calculate the piezoelectric coefficient of a normal PZT film with c+/c- domain under different electric field for comparison. (2) Will the Landauer's paradox influence the piezoelectric response? Since it is well known that the domain wall itself can have large piezoresponse due to low moving barrier, a larger barrier would indicate a lower response for the DWs which might reduce the actual piezoelectric coefficient.

Rebuttal letter to referees (NCOMMS-23-22122B)

We would like to thank the Referees for their helpful questions and comments. In the process of answering them and revising the text, our manuscript has been improved. Below, we address all their concerns in detail. The resulting modifications to the manuscript clarify some of the most important points of our work. The revisions and additions made in the manuscript are highlighted by red colored font in the marked-up version.

For each Referee, we reply to each question raised by first quoting the question followed by our response.

Response to Referee 1

Comment 1: The authors have successfully and carefully addressed most of the points I raised.

Reply 1: We thank this Referee for mentioning that “The authors have successfully and carefully addressed most of the points I raised”. Below the questions and comments posed by the Referee are fully addressed.

Comment 2: After reading the authors' response, my main concern is that the authors claim that the results depend on the cooling rate employed, and that they have obtained the phase diagrams in this study using an extremely fast cooling.

Rapid quenching typically results in the system getting trapped in local minima of the free energy, which most of the times does not correspond to the experimental situation (in which the system shows the structure corresponding to the *global* minimum of the free energy). If the phases described in the manuscript are not the actual global minima of the system at a given temperature, then the relevance of this work would be much narrower, since it would be describing metastable phases of ferroelectric thin films (possibly experimentally accessible via extremely fast cooling), but not truly stable states. In fact, in their response the authors mention that for slower cooling rates they obtained a different phase, stripes instead of labyrinths. A similar result was reported by some of the authors in Nature 577, 47, where some of the phases in the simulation were only obtained under very fast annealing.

Could the authors elaborate on the justification for using an extremely fast cooling rate? In particular, can the authors compare the free energies of the labyrinth domain obtained with the very fast quenching (results of the main text) and the stripe domain state obtained with the slower quenching (mentioned in the report) at the same (final) temperature? I assume, although it is not completely clear to me from the authors' response in Reply 4, that this comparison is at zero field. Then, if the free energy of the labyrinth is larger than that of the stripes at zero field, that would mean that the authors' model predicts the more ordered phase (stripes) to be more stable than the more

disordered phase (labyrinth), painting a very different picture from that described in the present work, which would deserve to be mentioned in the text.

Reply 2: We thank the Referee's comment here. In fact, labyrinths have been observed by measurements [see, e.g., Nature Communications, 11, 5779 (2020)], which implies that our predictions are relevant and can be experimentally checked. The observation of these labyrinths is likely due to the fact that the energy difference between labyrinths and stripes is very small. For instance, at 20K with zero field (classical Monte Carlo (CMC)), we numerically found that the stripe pattern just has a 1.14 meV/formula (5-atom in the unit cell) lower internal energy than that of labyrinths. Under zero field, quenching (fast cooling) typically provides the labyrinths (metastable) states in the CMC simulations while annealing (slow cooling) generally yields the (ground state) stripes. However, it is also interesting to realize that conducting PI-QMC calculations can also make the system adopting labyrinths even under annealing, implying that quantum fluctuations can overcome the small barrier between stripes and labyrinths at low temperature.

Comment 3: A smaller concern of mine is that of the bimeronic character of phase III and of the definition of the bimeron itself. In the references pointed out by the authors in their response (Reply 5), bimeronic topological structures are depicted and described. Still, in one of the references [Nature Communications, 11, 5779] some of the authors are also authors of the present manuscript and deal with essentially the same structure as in this work - I assume the reasoning behind calling these objects bimerons has probably been inherited from that work. In another reference [Review of Modern Physics, 95, 025001] that previous work is cited and partly reproduced, but no additional description or insight on the bimeronic structure of this phase is given. In the other referenced article in the response [Phys. Rev. B 83, 100408(R)], the structure depicted is homeomorphic to a (Bloch) skyrmion.

In other articles bimerons are described as objects with an in-plane periphery and a bimodal out-of-plane core:

- Phys. Rev. B 99, 060407(R) (2019)
- npj Comput Mater 6, 169 (2020)
- Appl. Phys. Lett. 118, 052411 (2021)
- Opt. Lett. 46, 3737-3740 (2021)
- Physics Reports, 895, 1 (2021)

In all these reports bimerons are shown as a bimodal distortion at the core perpendicular to its periphery (where the periphery should be extensible to the infinity). However, this does not correspond to the object that the authors describe in Fig. S2 (blue dashed line); instead the structure in the present work resembles a Néel-type skyrmion rather than a bimeron.

Then, and in order to clarify this issue, I would request from the authors to show the computed topological charge (with an explicit definition of the computed quantity)

from which they deduce that the object enclosed by the blue dashed lines in Fig. S2 is a bimeron. (Note also that the skyrmionic number of a bimeron may be equal to 1, see e.g. Physics Reports, 895, 1, so computing the skyrmionic number may not be sufficient to distinguish between a bimeron and a skyrmion).

Otherwise, the findings of the article are still valid - but I would then insist on using a different name for the objects found in phase III.

Reply 3: We greatly appreciate the detailed explanation provided by Referee 1. In our opinion, a “bimeron” can be simply defined as a pair of linked merons. This definition fits all the bimeron structures described in the references above.

For instance, below we schematically show a bimeron pattern discussed in Phys. Rev. B 83, 100408(R) (2011).

Fig. R1 Bimeron pattern from Phys. Rev. B 83, 100408(R) (2011).

Here, the core of each meron is highlighted by a yellow background, while the two Bloch walls linking the merons are highlighted by thick blue lines. This structure is indeed homeomorphic to a Bloch skyrmion.

The exact same picture is seen for another bimeron realization described in Phys. Rev. B 99, 060407(R) (2019). Below we provide an illustration of this structure with a slightly increased distance between merons:

Fig. R2 Bimeron pattern from Phys. Rev. B 99, 060407(R) (2019) with an increased distance between merons.

Once again, the meron cores are highlighted in yellow, while the domain walls connecting the merons are shown by blue lines. This time the walls feature Néel rather than Bloch rotations. Overall, this structure is homeomorphic to a Néel skyrmion.

Indeed, a rotation of all vectors in Fig. R2 around the vertical y axis by 90° results in the following structure:

Fig. R3 Bimeron structure resulting from a 90° rotation of all vectors in figure R2 around the vertical y axis.

which is clearly homeomorphic to a Néel skyrmion.

Additionally, one can note that the bimeron structure shown in Fig. R3 transforms into the bimeron pattern in Fig. R1 when all vectors are, once again, rotated by 90° around the out-of-plane z axis.

Therefore, all three bimeron patterns (Figs. R1-3) are homeomorphic one to the other and each is homeomorphic to a skyrmion. The distinction between these bimerons and a skyrmion is purely geometric – the Skyrmion charge density in the former is broken into two parts, each corresponding to a $\frac{1}{2}$ charged meron.

In our work, the bimerons have a structure equivalent to the one shown in Fig. R3 and we do not see a reason to introduce yet another term to refer to these objects. The classification of different polar and spin patterns is already quite complex and, in our opinion, introducing more terms would only encumber it more.

At the same time, we absolutely agree with the Referee in that providing the computed skyrmion charge density for the bimerons would be beneficial. We numerically computed skyrmion charge density based on the formula $q = \frac{1}{4\pi} \mathbf{u} \cdot (\partial_x \mathbf{u} \times \partial_y \mathbf{u})$ adapted to vector fields on the lattice with the Berg and Luscher approach [Nucl. Phys. B 190, 412 (1981)] for each unit cell. The dipolar structure and skyrmion charge density for the same plane (the 4th layer) are shown in Figs. R4a and R4b (see below). One can clearly see that the skyrmion charge density is broken into two parts, each corresponding to a $\frac{1}{2}$ charged meron. Now, we have provided the corresponding data in the Supplemental Information.

Fig. R4 The dipolar structure (a) and skyrmion charge density (b) for the $z=4$ plane of the supercell. The bimeron pattern is indicated by a gray rectangle. One can clearly see that the skyrmion charge density (ρ_{sk}) is broken into two parts (blue rectangles), each corresponding to a $\frac{1}{2}$ charged meron.

Comment 4: List of small typos:

Line 48: "quantitively" -> "quantitatively"

Line 114: the reference to Fig. S1 should be to Fig. S2 instead.

Reply 4: We thank the Referee for pointing out these typos. In the new version, we have corrected them and highlighted in red these corrections.

Response to Referee 2

Comment 1: The authors have addressed all my comments, I'm happy to recommend for publication.

Reply 1: We thank this Referee for mentioning that "The authors have addressed all my comments, I'm happy to recommend for publication". Below the questions and comments posed by the Referee are fully addressed.

Comment 2: Would be valuable if the authors can calculate the piezoelectric coefficient of a normal PZT film with $c+/c-$ domain under different electric field for comparison.

Reply 2: We thank the Referee for this suggestion. We thus calculated the piezoelectric coefficient when the initial (i.e., at zero field) state is formed by stripes (that are the $c+/c-$ domains) for PZT film under dc electric field within CMC simulations. The results are shown in Fig. R5b. For comparison, Fig. R5a displays the piezoelectric

coefficient when the initial state (i.e., at zero field) is made of labyrinths for PZT film under dc electric field within CMC simulations. One can see that for the “normal” PZT film with $c+/c-$ domain (stripes), with low electric fields, the piezoelectric coefficient nicely increases linearly with small oscillations. This is because low electric fields do not change the shape of the stripe pattern, only shrinking the width of stripes. For very large electric fields, both types of CMC simulations yield monodomain (dipoles point to the electric field direction), thus providing the same behavior for the piezoelectric coefficient. All in all, the piezoelectric coefficients of Figs R5a and R5b are rather similar for the different electric fields.

Fig. R5 (a) The initial state (zero electric field) is made by labyrinths. The resulting piezoelectric coefficient (d_{33}) as a function of electric field from CMC. (b) The initial state (zero electric field) is formed by stripes. The resulting piezoelectric coefficient (d_{33}) as a function of electric field from CMC.

Comment 3: Will the Landauer's paradox influence the piezoelectric response? Since it is well known that the domain wall itself can have large piezoresponse due to low moving barrier, a larger barrier would indicate a lower response for the DWs which might reduce the actual piezoelectric coefficient.

Reply 3: Defects (that are at the origin of the Landauer's paradox) indeed likely affect the piezoelectric response, as well as most other physical responses. For instance, [Physical Review B 75, 014111 (2007)] predicts the effect of lead vacancies on dielectric response of a perovskite material and provides a microscopic insight into such effect.

The followings are the revisions we made in the main text and supporting information

1, For the response of comment 3 of Referee 1, we have added the following Figures as Fig. S2 in the supporting information of part II and highlighted the changes with red color.

2, For the response of comment 4 of Referee 1, we have changed “*quantitively*” to be “*quantitatively*” and changed the “Fig. S1” to be “Fig. S2” in the paragraph 3 of page 2 of main text and highlighted the changes with red color.

3, For the response of comment 3 of Referee 2, we have added “*Note that the piezoelectric response or any other physical response can be affected by the presence of structural defects, as, e.g., shown in Ref.⁵⁹*” in the paragraph 1 of page15 of main text and highlighted the changes with red color.

REVIEWERS' COMMENTS

Reviewer #1 (Remarks to the Author):

The authors have addressed my concerns making valid points.

The discussion about the bimerons has helped me understand the nomenclature chosen for phase III. I share with the authors the spirit of not introducing yet another name for a polar pattern unless it is genuinely distinct. Hence I suggest to refer to the objects in phase III simply as skyrmions (or at most "elongated skyrmions"), since the fact that they are elongated in one direction does not affect their topological character and describes the dipole pattern with accuracy. In any case, I believe the authors made a reasonable point for keeping the name bimeron for the structure they find.

Therefore, I recommend that the article be published (regardless of the name the authors finally employ for phase III).

Reviewer #2 (Remarks to the Author):

The authors have addressed my comments. one minor comment: The d_{33} value for PZT is one order of magnitude than the experimental value.

Rebuttal letter to referees (NCOMMS-23-22122B)

We would like to thank the Referees for their helpful questions and comments. In the process of answering them and revising the text, our manuscript has been improved. Below, we address all their concerns in detail.

For each Referee, we reply to each question raised by first quoting the question followed by our response.

Response to Referee 1

Comment 1: The authors have addressed my concerns making valid points. The discussion about the bimerons has helped me understand the nomenclature chosen for phase III. I share with the authors the spirit of not introducing yet another name for a polar pattern unless it is genuinely distinct. Hence I suggest to refer to the objects in phase III simply as skyrmions (or at most "elongated skyrmions"), since the fact that they are elongated in one direction does not affect their topological character and describes the dipole pattern with accuracy. In any case, I believe the authors made a reasonable point for keeping the name bimeron for the structure they find. Therefore, I recommend that the article be published (regardless of the name the authors finally employ for phase III).

Reply 1: We thank the Referee for recommending the publication of our paper in Nature Communications. The classification of different polar and spin patterns is already quite complex and, in our opinion, introducing more terms would only encumber it more. Hence, we indeed prefer to continue to refer to bimeron for phase III

Response to Referee 2

Comment 1: The authors have addressed my comments.

Reply 1: We appreciate that the Referee stated, "The authors have addressed my comments". Below, the questions and comments posed by the Referee are fully addressed.

Comment 2: One minor comment: The d_{33} value for PZT is one order of magnitude than the experimental value.

Reply 2: Two reasons may be invoked to explain this discrepancy. First of all, the experimental d_{33} value is obtained at room temperature, whereas our d_{33} is calculated at a rather low temperature of 20 K. It is well documented that decreasing the temperature below the Curie temperature dramatically reduces the piezoelectric

coefficients in PZT systems. Secondly, experiments are conducted on monodomains, whereas our computations are performed on multidomains.